

# Functionalized nanoparticles crossing the brain–blood barrier to target glioma cells

Yongyan Wu[1,*], Yufeng Qian[2,*], Wei Peng[3] and Xuchen Qi[1,2]

[1] Department of Neurosurgery, Sir Run Run Shaw Hospital, Zhejiang University School of Medicine, Hangzhou, Zhejiang, People's Republic of China
[2] Department of Neurosurgery, Shaoxing People's Hospital, Shaoxing, Zhejiang, People's Republic of China
[3] Medical Research Center, Shaoxing People's Hospital, Shaoxing, Zhejiang Province, People's Republic of China
[*] These authors contributed equally to this work.

## ABSTRACT

Glioma is the most common tumor of the central nervous system (CNS), with a 5-year survival rate of <35%. Drug therapy, such as chemotherapeutic and immunotherapeutic agents, remains one of the main treatment modalities for glioma, including temozolomide, doxorubicin, bortezomib, cabazitaxel, dihydroartemisinin, immune checkpoint inhibitors, as well as other approaches such as siRNA, ferroptosis induction, *etc*. However, the filter function of the blood-brain barrier (BBB) reduces the amount of drugs needed to effectively target CNS tumors, making it one of the main reasons for poor drug efficacies in glioma. Thus, finding a suitable drug delivery platform that can cross the BBB, increase drug aggregation and retainment in tumoral areas and avoid accumulation in non-targeted areas remains an unsolved challenge in glioma drug therapy. An ideal drug delivery system for glioma therapy should have the following features: (1) prolonged drug life in circulation and effective penetration through the BBB; (2) adequate accumulation within the tumor (3) controlled-drug release modulation; (4) good clearance from the body without significant toxicity and immunogenicity, etc. In this regard, due to their unique structural features, nanocarriers can effectively span the BBB and target glioma cells through surface functionalization, providing a new and effective strategy for drug delivery. In this article, we discuss the characteristics and pathways of different nanocarriers for crossing the BBB and targeting glioma by listing different materials for drug delivery platforms, including lipid materials, polymers, nanocrystals, inorganic nanomaterials, *etc*.

Corresponding authors
Xuchen Qi, qixuchen@zju.edu.cn
Wei Peng, pengweiaqw@163.com

## INTRODUCTION

Gliomas are the most common primary brain tumors. They account for 81% of all central nervous system (CNS) malignant tumors (*Ghouzlani et al., 2021*; *Xu et al., 2020b*), and have a relatively poor 5-year overall survival at <35% (*Lapointe, Perry & Butowski, 2018*). In recent years, immunotherapy, including immune checkpoint inhibitors, DC vaccines, oncolytic viral, CAR-T cell therapy, *etc.*, was shown to have promising effects on glioma

(*Carreno et al., 2015*; *Mathewson et al., 2021*; *Xu et al., 2020b*). However, despite the potential of chemotherapeutic and immunotherapeutic drugs in suppressing glioma growth and progression, their effectiveness is greatly reduced by biological barriers in the body (*Hao et al., 2021*).

The blood–brain barrier (BBB) serves as a protective barrier for the brain by separating circulating blood from the extracellular fluid of the CNS to prevent the transition of substances such as hormones, neurotransmitters or neurotoxins from the blood to the CNS (*Betzer et al., 2017*). Brain endothelial cells and their interacting astrocytes, pericytes, microglia, neurons, mast cells and circulating immune cells crosstalk to form the neurovascular unit (NVU), which helps refine the function of the BBB and maintain barrier physiological properties (*Banks, 2016*).

During tumor development, the normal structure of the BBB is gradually destroyed, resulting in the formation of the blood–brain tumor barrier (BBTB) (*Karim et al., 2016*). BBTB is characterized by abnormal blood vessel formation and altered molecular transport. The changes in BBTB are mainly in structure, permeability, molecular transporters and receptors (shown in Fig. 1):

(1) Structure: Glioma BBTB is characterized by abnormal blood vessel formation and increased permeability due to damage caused by the tumor. Tight junctions (TJs) are a critical component of the BBB and are responsible for regulating the paracellular transport of molecules across the endothelial cells that form the barrier (*Yang et al., 2019*). In glioma BBTB, the TJs are disrupted, leading to increased permeability of the barrier and altered molecular transport across the BBB (*Mojarad-Jabali et al., 2021*; *Rathi et al., 2022*).

(2) Permeability: Glioma BBTB is characterized by increased permeability, allowing molecules to move more freely between the blood and the brain (*Li et al., 2020*), such as vascular endothelial growth factor (VEGF) (*Seyedmirzaei et al., 2021*), matrix metalloproteinases (MMPs) and cytokines such as IL-1 $\beta$ (*Xue et al., 2019*; *Zhou et al., 2019*).

(3) Molecular transporters and receptors: Glioma BBTB has altered expression of these transporters and receptors, which can lead to abnormal accumulation of certain molecules in the brain (*Quader, Kataoka & Cabral, 2022*). For example, efflux transporters are a group of proteins that are responsible for transporting various molecules and drugs out of cells, which play a critical role in maintaining the integrity and function of the BBB (*van Tellingen et al., 2015*). In glioma BBTB, the expression and function of efflux transporters can be altered such as P-glycoprotein (P-gp), breast cancer resistance protein (BCRP), and multidrug resistance protein 1 (MRP1), leading to decreased drug penetration into the brain and resistance to chemotherapy (*Quader, Kataoka & Cabral, 2022*; *Veringa et al., 2013*), which have significant clinical implications for the treatment and management of gliomas.

While glioma BBTB is more permeable than BBB, it can still exhibit a high degree of heterogeneous permeability to drugs due to its role as a physical and metabolic barrier in the tumor microenvironment and can also contribute to the growth and spread of gliomas (*Arvanitis, Ferraro & Jain, 2020*). Therefore, identifying ways for the target delivery of therapeutic agents across the BBB is a pressing issue for glioma treatment.

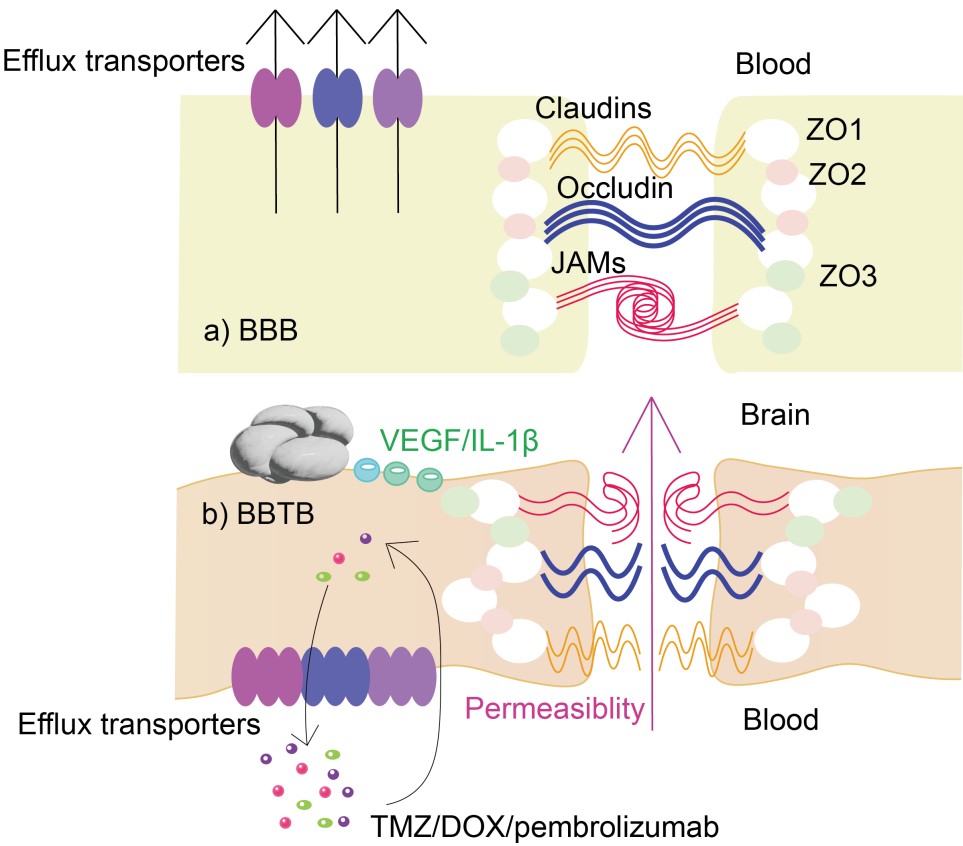

**Figure 1  Changes in the structure and function of glioma BBTB compared to BBB.** The figure demonstrates the structural and functional differences between (A) normal BBB and (B) glioma BBTB. In contrast to (A), glioma in (B) causes upregulation of the levels of VEGF, IL-1 and other substances, leading to disruption of the normal structure of the BBB, which in turn leads to increased permeability. Also, glioma BBTB in (B) causes alterations in efflux transporters, the upregulation of whose levels leads to more clearance of, for example, temozolomide, doxorubicin, and immune checkpoint inhibitors, resulting in a significant reduction in the effectiveness of drug therapy.

Nanotechnology fused intracranial drug delivery appears to be an effective strategy. Nanomedicine has yielded innovative products for targeted drug delivery, which has helped reduce drug toxicity, improve bioavailability, and more importantly, increase drug permeability through biological barriers (*Lu et al., 2021*).

In this article, we summarized the therapeutic drugs and molecular biological characteristics of glioma, the mechanism for BBB modeling and the current research progress of common nanomaterials as drug carriers for glioma.

## Why this review is needed and who it is intended for

Functionalized nanoparticles have emerged as efficient delivery platforms in glioma therapy. Recent work has demonstrated nanoparticles of various materials and their different functionalized targets. This review discusses and summarizes the new advances in functionalized nanoparticles in recent years and the respective advantages of different

functionalized nanocarriers. Our review will appeal to researchers interested in nanoparticles in the field of glioma therapy, as well as an overview of research on the combination of nano drug delivery system and tumor targeting in glioma therapy.

## SURVEY METHODOLOGY

Data were collected as previously described in *Pham et al. (2014)*. Specifically, we conducted a scoping review to scope a body of literature, and clarify concepts for this review using PubMed, with the last search conducted on 2023 (*Cazorla-Ortiz et al., 2020*; *Pham et al., 2014*). The search was performed in full-text journals, focusing on the drug delivery through BBB and tumor targeting of nanoparticles and their role in glioma therapy. The keywords used and their synonyms and variants and combinations of the words used for the search are as follows:

(1) Nanoparticles: nanoparticles; liposomes; solid lipid nanoparticles (SLNs); nanostructured lipid carriers (NLCs); polymer; micelles; biomaterial nanoparticles; extracellular vesicles (EVs); high-density lipoproteins (HDL); carbon nanoparticles; carbon nanotubes (CNT); carbon nanodots (CND); carbon nanosphere (CSP); gold nanoparticles (AuNPs); nanocrystals.

(2) Glioma targeting and BBB crossing: transposable protein receptor (TFR); low-density lipoprotein receptor (LDLR); vascular endothelial growth factor (VEGFR); epidermal growth factor receptor (EGFR); integrins; glucose transporter (GLUT); receptor-mediated transport (RMT); carrier-mediated transport (CMT); adsorption-mediated transport (AMT); P-glycoprotein (P-gp); multidrug resistance proteins (MRPs); convection-enhanced delivery (CED); passive targeting; active targeting.

(3) Therapy: glioma; invasion; migration; cell proliferation; angiogenesis; immune escape; immunosuppression; cell death; apoptosis; temozolomide (TMZ); bortezomib; cabazitaxel; immune checkpoint inhibitors; siRNA; signal transducer and activator of transcription 3 (STAT3); ferroptosis.

The words were merged *via* the Boolean operators 'AND' and 'OR' (*Shi et al., 2022c*). The initial search screened approximately 400 relevant articles written in English that could be useful for this review.

## RECEPTORS SPECIFICALLY EXPRESSED ON THE SURFACE OF GLIOMA CELLS

With the research progress on the molecular mechanism of glioma pathogenesis, a large amount of evidence has shown that changes in glioma surface-related receptors contribute to the aggressive biological behavior of glioma (*Saadeh, Mahfouz & Assi, 2018*; *Xun et al., 2021*). Therefore, targeting the receptors has become a new strategy for the treatment of glioma (*Yang et al., 2022*).

### Transposable protein receptor (TFR)

TFR is highly expressed in brain microvascular endothelial cells and glioma cells (*Kang et al., 2020*). Some examples of potential ligands utilizing TfR as a target include TF, L-/D-T7 peptide, OX26 mAb, *etc.* (*Béduneau et al., 2008*; *Guan et al., 2021*; *Tang et al., 2019a*).

### Lactoferrin receptor (LfR)

Lactoferrin (Lf) is a naturally occurring iron-binding protein that has been shown to specifically bind to the Lactoferrin receptor, which is overexpressed in glioma (*Song et al., 2017*). Lactoferrin can be used as a targeting agent for drug delivery to gliomas, such as the study that paclitaxel was conjugated to lactoferrin (*Miao et al., 2014*) and another study of a lactoferrin-hyaluronic acid (HA) dual-targeting drug delivery system (*Yin et al., 2016*).

### Low-density lipoprotein receptor (LDLR)

The LDLR family includes low-density lipoprotein receptor-related proteins-1 (LRP-1), LRP-1B, LRP-2, Apo-E receptor 2, MEGF7 and LDLR, overexpressed in BBB and glioma cells (*Pawar et al., 2021*; *Wei et al., 2021*). Some targeting ligands have been proven to be used in LDLR targeted therapy of glioma such as LDL particles, LDLR antibodies, Angiopep 2, Apo-B, Apo-E and some certain peptides (*He et al., 2021*; *Pawar et al., 2021*; *Zhang et al., 2013*).

### EGFR

EGFR belongs to the ErbB receptor family widely overexpressed in glioma cells and can be activated through the upregulation of kinases such as JAK, p85MAPK, *etc.* (*Chistiakov, Chekhonin & Chekhonin, 2017*; *Sabbah, Hajjo & Sweidan, 2020*). EGFR variant III (EGFRvIII), a deletion mutant of EGFR, can activate downstream pathways to increase tumor aggressiveness. However, the therapeutic effects of many antibody drugs targeting EGFR are limited against the EGFRvIII pathway (*Greenall et al., 2019*; *Mao et al., 2017*; *Saadeh, Mahfouz & Assi, 2018*).

### VEGFR

VEGFR, such as VEGFR-1 (flt-1), VEGFR-2 (flt-1/KOR) and VEGFR-3 (flt-4), are tyrosine kinase receptors located on the surface of vascular endothelial cells, and studies have shown that VEGFR is highly expressed in both gliomas and vascular endothelial cells (*Cheng, Liu & Rao, 2021*; *Ying et al., 2022*).

### Integrin protein

Integrins are transmembrane glycoprotein receptors with cell adhesion and signaling protein functions (*Slack et al., 2022*). They include $\alpha v \beta 3$ and $\alpha v \beta 5$, which are overexpressed on glioma and tumor neovascular endothelial cells as well as tumor-associated macrophages (*Qi et al., 2021*; *Wei et al., 2019*).

### Glucose transporter (GLUT)

GLUTs are carrier-mediated transporters through which glucose is transported through the BBB. Tumor cells usually have an overexpression of glucose transporters, especially GLUT-1 and GLUT-3 due to their glucose requirement (*Fu et al., 2019*; *Fu et al., 2019b*; *Jia et al., 2022*; *Li et al., 2016*).

### Folate receptor (FR)

FR is a glycoprotein on the cell surface that recognizes folic acid and mediates its uptake. It has been found that FR is highly expressed in many cancers, including about 90% in

brain tumors. In response to the abnormal high expression of FR in glioma, BBB and BBTB cells, folic acid was used as a target agent for drug delivery to glioma, binding directly to the drug or binding it into nanoparticles (*Shi et al., 2022b*).

### Neuropilin-1 (NRP-1)

NRP-1 is a transmembrane protein that is overexpressed on the surface of glioma cells, which is involved in multiple biological processes, including cell migration, angiogenesis, and immune regulation (*Jia et al., 2018*). NRP-1 has been identified as a potential target for glioma therapy (*Chen et al., 2015*; *Gries et al., 2020*).

## STRATEGIES TO CROSS/BYPASS BBB

Despite the shielding effects of the BBB, some small molecules, such as glucose and hydrophobic molecules with molecular weights less than 500 Da, and certain cells, such as monocytes, macrophages and neutrophils, can still be selectively transported through the brain (*Xie et al., 2019*). However, it should be noted that not the smaller the molecule, the easier it is to pass the BBB (*Kadry, Noorani & Cucullo, 2020*). The mechanisms of BBB crossing under physiological conditions (illustrated in Fig. 2) include (1) receptor-mediated transport (RMT); (2) paracellular pathway; (3) passive diffusion; (4) carrier-mediated transport; (5) cell-mediated transport; and (6) adsorption-mediated transport (AMT) (*Hervé, Ghinea & Scherrmann, 2008*; *Malcor et al., 2012*; *Reddy et al., 2021*; *Ruano-Salguero & Lee, 2020*; *Tang et al., 2019b*).

Efflux transporters play a crucial role in the blood–brain barrier (BBB) by actively pumping various substances out of brain tissue into the blood circulation such as P-glycoprotein (P-gp) and breast cancer-related proteins (BCRP), which belong to the ATP-binding cassette (ABC) transporter family (*Shuai et al., 2022*). P-gp, encoded by the ABCB1 gene, is highly expressed in the luminal (blood-facing) membrane of brain capillary endothelial cells that form the BBB (*Han, Gao & Mao, 2018*). It can recognize and transport a wide range of structurally diverse substances, including chemotherapeutic drugs, opioids, immunosuppressants, and toxins (*Zhang et al., 2022a*). BCRP, encoded by the ABCG2 gene, is another important efflux transporter in the BBB (*Han, Gao & Mao, 2018*). Similar to P-gp, BCRP has a broad substrate specificity and can transport various drugs, such as anticancer agents, antibiotics, and antiviral drugs (*Peña Solórzano et al., 2017*). The high expression of efflux transporters can actively pump therapeutic drugs out of the brain, resulting in lower drug concentrations at the target site (*Gomez-Zepeda et al., 2019*). However, efflux transporters can be utilized in glioma treatment to enhance drug delivery to the tumor site. Here are two main strategies: (1) Prodrug: A prodrug is an inactive or less active form of a drug that can be converted into its active form at the target site (*Majumdar, Duvvuri & Mitra, 2004*). Prodrugs that are not recognized or efficiently transported by these transporters as some studies have explored (*Teng et al., 2022*). (2) Efflux transporter inhibitors: efflux transporter inhibitors, such as probenecid (PRB), can be co-administered with therapeutic drugs to block the activity of efflux transporters (*Huttunen, Gynther & Huttunen, 2018*), resulting in increased drug concentrations at the

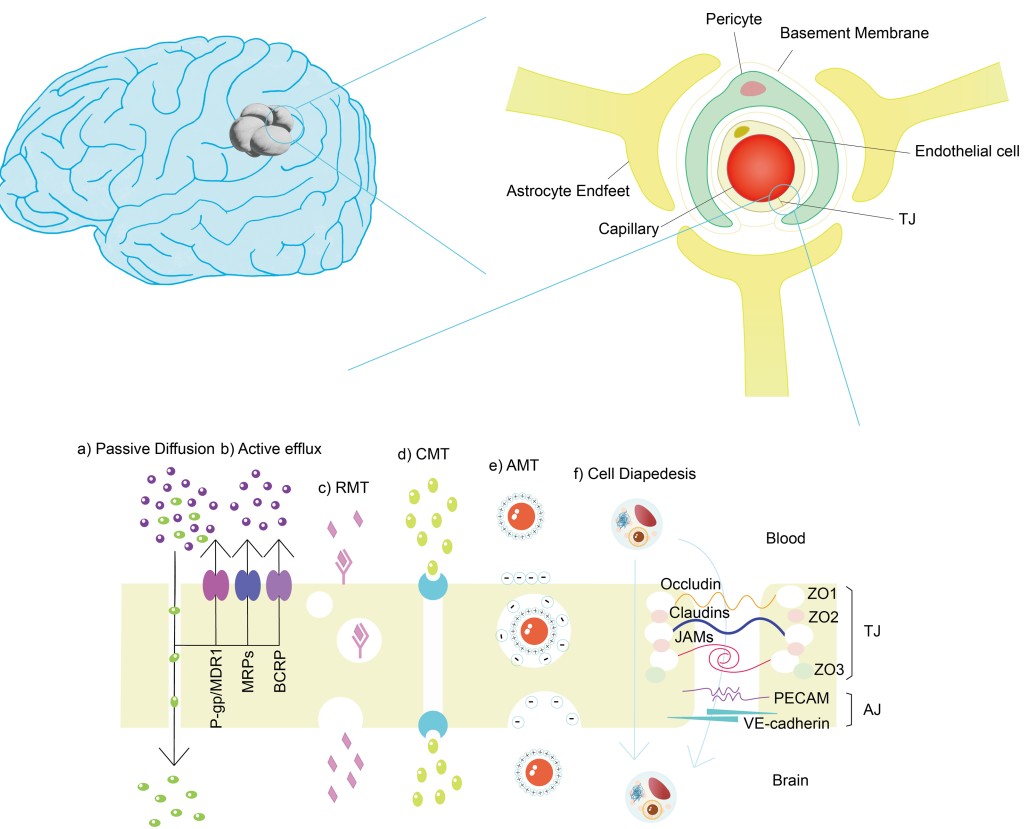

**Figure 2** **Structure of BBB and mechanisms of cross BBB.** (A) Passive diffusion: small lipophilic molecules with molecular weight less than 400–500 Da and less than nine hydrogen bonds. (B) Active efflux: drugs, drug conjugates, xenobiotics, nucleosides. (C) RMT: arginine-vasopressin, transferrin, leptin, insulin, apolipoproteins, amyloid-$\beta$, glycosylated proteins, *etc.* (d) CMT: amino acids, carbohydrates, monocarboxylic acids, fatty acids, hormones, nucleotides, organic anions, amines, choline, vitamins, *etc.* (e) AMT: large peptides or proteins, such as IgG. (f) Cell diapedesis: including transcellular and paracellular pathway, such as leukocyte.

target site (*Choi & Yu, 2014*). However, more work is still needed to improve in terms of transporter recognition, selectivity, and BBB penetration.

The barrier effect of the BBB can be also bypassed by direct administration through the intracranial tumor area. Although the injection of intra-tumor drugs seems an effective way, it is limited in practical application due to many factors, such as bleeding, increased risk of intracranial infection, and limited diffusion range of intra-tumor drugs (*Janjua et al., 2021*). Convection-enhanced delivery (CED) uses a minimally invasive procedure to place a microcatheter inside a tumor (*Mehta, Sonabend & Bruce, 2017*). However, the effect is determined by complex factors such as backflow, air bubbles, tissue edema, pressure gradient, *etc.* (*Mehta, Sonabend & Bruce, 2017*). Implantable drug-loaded polymers such as wafers, gels or microchips are another option (*Yu et al., 2019*). Taking Gliadel® for an example, it has been shown to prolong OS (*Zhao et al., 2019*) but also showed an increased risk of postoperative cerebrospinal fluid leakage and infection (*Chakroun et al., 2018*).

The disruption of the BBB also provides another option for intracranial drug delivery. Mannitol and bradykinin can act on endothelial cells to contract them or on B2 receptors in endothelial cells, causing a transient opening of TJs and enhancing BBB ground permeability (*Karim et al., 2016*). The BBB can also be disrupted by ultrasound or photothermal effects to increase drug delivery and prolong drugs' half-life (*Umlauf & Shusta, 2019*; *Yan et al., 2022*). However, the timing of the induction of BBB disruption must be controlled, as this might increase the risk of intracranial infection and accumulation of toxins (*Janjua et al., 2021*).

# NANOPARTICLES (NPs)

NPs that can be used for drug delivery include inorganic nanomaterials, dendrimers, polymers, micelles, liposomes, nanocrystals, *etc.* (*Li et al., 2020*). Certain types NPs can cross the BBB and have shown promising potential for glioma treatment (*Qiao et al., 2022*). By designing the optimal size, shape and surface features, NPs can cross the BBB more easily, carry a variety of hydrophilic and hydrophobic drugs, and protect these drugs from degradation as illustrated in Fig. 3. On the other hand, NPs use various biological reactions such as receptor–ligand and antigen-antibody interactions to modify their surfaces to conjugate with specific molecules and achieve functionality for the targeted delivery *in vivo* (*Amiri et al., 2021*; *Sim et al., 2020*).

NPs for glioma drug delivery are generally designed in the following two modes: glioma microenvironment responsive NPs and NPs modified with specific cell membrane surfaces (*Li et al., 2022*). Glioma microenvironment responsive NPs are engineered to respond to specific features of the glioma microenvironment, such as low pH in glioma cells (*He et al., 2021*), high levels of reactive oxygen species (ROS) (*Chen et al., 2022*; *Zhang et al., 2022b*), and overexpression of certain proteins such as EGFR (*Mao et al., 2017*). By targeting these specific features, they can increase the efficacy of drug delivery to glioma cells while minimizing side effects on healthy cells (*Xu et al., 2020a*). NPs for glioma treatment modified with specific cell membrane surfaces is a promising approach in targeted drug delivery, which are coated with the membranes of cells that are specifically chosen for their affinity for glioma cells while reducing damage to healthy cells. The specific cell membranes used for coating nanoparticles can vary, but often include those from immune cells such as macrophages, red blood cells, platelets, and cancer cells themselves which have been shown to preferentially bind to glioma cells (*Shi et al., 2022a*; *Cui et al., 2020*; *Wang et al., 2023*; *Wu et al., 2021*).

## Liposomes nanoparticles (LNPs)

LNPs have been used to treat various diseases as a promising delivery carrier such as liposomes, solid lipid nanoparticles (SLNs) and nanostructured lipid carriers (NLCs).

Liposomes consist of a lipid bilayer and a core of hydrophilic vesicles. Their main components are lipids and fatty acids, which are widely found in biological membranes. They have excellent biocompatibility, allowing both hydrophilic drugs to be loaded in the core and lipid-soluble drugs to be loaded between the lipid bilayers (*Jagaran & Singh, 2022*; *Large et al., 2021*). The design of PTX-loaded Rg3-LPs on C6 glioma cells was shown
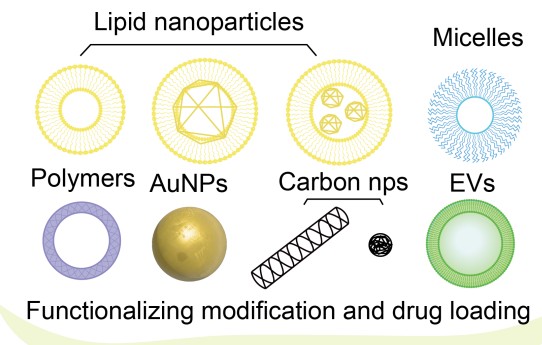

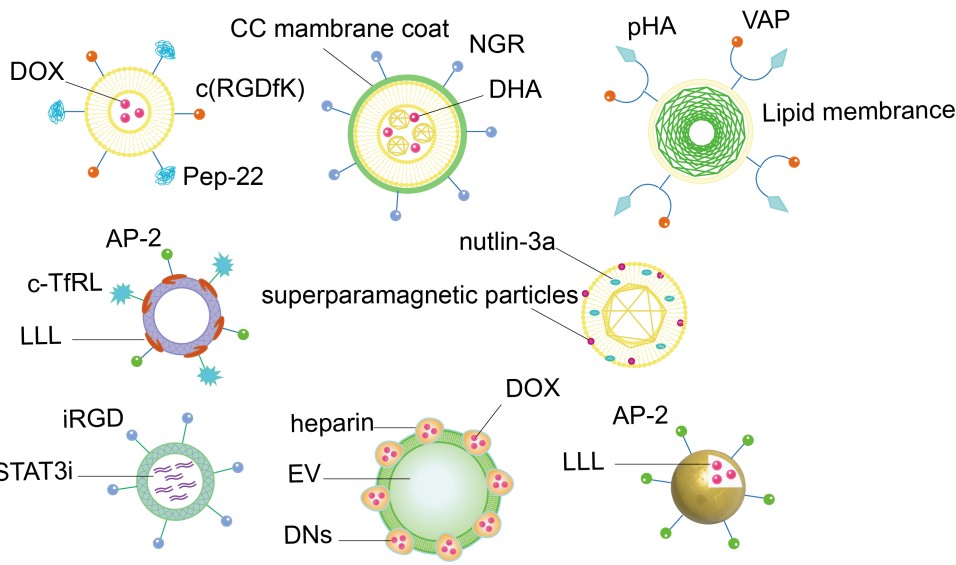

**Figure 3  Nanoparticles with functionalizing modification and drug loading.** The figure shows NPs composed of different materials, such as lipid-based NPs, micelles, polymer NPs, AuNPs, carbon structured NPs, biomaterial NPs, which are functionally modified by different ligands and materials, and loaded with drugs to achieve a functionalized nanodrug delivery platform that can dual-target BBB and glioma cells.

not only improved the permeability to the BBB but also enhanced the targeting and diffusion ability of glioma *in vivo* (*Fanciullino, Ciccolini & Milano, 2021*; *Zhu et al., 2021*).

SLNs and NLCs are new types of nano-drug-loading system developed over the past decade, which have the advantages of good biocompatibility, controlled drug release and ease of large-scale industrial production as some researches have shown(*Chang et al., 2021*; *Paliwal et al., 2020*).

## Polymeric nanoparticles

Polymeric nanoparticles can be defined as colloidal systems, self-assembled from amphiphilic blocks, with a core–shell structure that facilitates the encapsulation and sustained release of hydrophobic drugs (*Mukherjee et al., 2019*). It was found that by
adjusting the size of polymer nanoparticles, the permeability and EPR effects of nanomaterials could be effectively enhanced to significantly increase the effective delivery of anti-tumor drugs. *Gao, Chen & Wu (2021)* designed a small particle-size polymer loaded with adriamycin can significantly inhibit gliomas' growth and greatly enhance their sensitivity to chemotherapy drugs.

## Micelles

Polymeric micelles are polymeric nanoparticles with good biocompatibility, which can passively target the tumor area through the EPR effect (*Guo et al., 2019*; *Paranthaman et al., 2022*). Micelles can be made pH-sensitive by exploiting the pH properties of tumor tissues, which is one of the main strategies being investigated (*Wang et al., 2018*; *Zhou et al., 2018*). *Jiang et al. (2021)* designed novel non-toxic cation-free siRNA micelles that could have both siRNA knockdown effects on STAT3 and carry small molecule drugs, such as temozolomide (TMZ, Temodal®, Temodar®), to successfully reach the glioma region for synergistic therapeutic effects.

## Biomaterial-based NPs

Extracellular vesicles (EVs) represent a natural carrier system that has a wide range of endogenous marker molecules on its surface, eliminating the need for additional targeting molecular modifications, avoiding the *in vivo* pathway and lysosomal degradation, evading the immune system and delivering the drug directly to the cells (*Balakrishnan et al., 2020*; *Rufino-Ramos et al., 2017*) as proved by *Jia et al. (2018)*.

High-density lipoproteins (HDL) could be another promising material for drug delivery (*Huang & Mahley, 2014*). *Cui et al. (2018)* used T7 and dA7R modified on natural HDL to target glioma through the BBB, which effectively delivered 10-hydroxycamptocampin (HCPT) to the glioma, which has shown a stronger anti-glioma therapeutic effect.

## Carbon NPs

Carbon NPS are materials composed mainly of carbon with a size of ≤100 nm (*Lewinski, Colvin & Drezek, 2008*). Carbon nanotubes (CNTs), graphene, carbon nanodots (CND) as well as carbon nanosphere (CSP) are relatively common among carbon-based nanoparticles (*Mocci et al., 2022*; *Modugno et al., 2015*; *Selvi et al., 2012*). *Grudzinski et al. (2014)* used carbon and iron to develop carbon-coated iron nanoparticles (CEINs) to treat glioma and reported that it was a good carrier and highly toxic to glioma.

## Gold NPs

Gold nanoparticles (AuNPs) are an alternative choice for drug delivery as carriers because of their inertness, high biocompatibility, relative ease of synthesis and functionalization (*Martínez-Rovira, Seksek & Yousef, 2019*). In a study, *Chandra Kaushik et al. (2019)* demonstrated that anti-EGFR-iRGD decorated erythrocyte membrane-derived NP could synergistically inhibit glioma EGFR with AuNP.

## Nanocrystals

Nanocrystals are amorphous or crystalline in nature produced by nanosizing pure drug particles (*Sverdlov Arzi & Sosnik, 2018*). The process and scale of nanocrystal production

have also made considerable progress with the improvement of the process (*Fontana et al., 2018*). Fusing other materials or technologies with nanocrystals is an efficient strategie to enhance their therapeutic effects (*Lin et al., 2014*). *Chai et al. (2019)* developed a targeted drug delivery system in which the targeted peptide-modified cell membrane was coated with drug nanocrystals, which greatly enhanced the retention and therapeutic effect of the drug in glioma.

## DRUGS TARGETING GLIOMA

The fundamental purpose of nanoparticle design is to serve as a carrier platform for smooth drug delivery through the BBB and to the tumor region. It can be generally divided into two mechanisms: passive and active targeting, the application of which is not conflicting (*Li et al., 2020*). Comparatively, the active targeting mechanism can achieve the fundamental role of the carrier more efficiently (*Yang et al., 2022*).

### Passive targeting

Passive targeting drug delivery is primarily based on the "enhanced penetration and retention effect" (EPR effect) of tumor tissue. At the same time, in advanced stages of glioma, passively targeted nanoparticles can even enter the tumor region through the interstitial space of endothelial cells (*Shi et al., 2020*). The most common drug delivery system using passive targeting is liposome mainly through the EPR effect (*Shi et al., 2020*). *Lu et al. (2021)* used lipoproteins in plasma as targeting factors to avoid liposomal drug-induced immune responses and exploit the properties of lipoproteins to cross the blood–brain barrier.

### Receptor-mediated active targeting

Receptor-mediated active targeting drug delivery platform exploits differences in receptor or antigen expression on the surface of tumor cells and normal cells to achieve higher trans-BBB transport capacity and tumor targeting (*Li et al., 2020*). The most common approach is the functionalized modification of the nanoparticle surface using different ligands (*Ghezzi et al., 2021*; *Mojarad-Jabali et al., 2021*). We have listed some receptors highly expressed on endothelial cells can act as targets (illustrated in Table 1), some of which expressed on glioma cells can be modified to achieve active targeting capabilities.

### Protein/peptide-mediated active targeting

Protein-like ligands, including transferrin, insulin, integrin, *etc.*, and peptide-like ligands, such as cell crossing peptide (CCP), RGD peptide, *etc.*, have been linked to these ligands to allow the vector to pass through the BBB by RMT effect.

For example, a nanoparticle, AuNP-AK-R, designed using cyclic RGD (cRGD)-modified AuNP, can recognize integrins $\alpha v \beta 3$ of glioma and actively target glioma cells (*Ruan et al., 2017*). A previous study reported using the integrin ligand c(RGDyK) as a targeting moiety and liposomes co-modified with DCDX constituting DCDX/c(RGDyK)-LS piggybacked on doxorubicin (DOX) for targeted drug delivery to glioma (*Wei et al., 2015*).

**Table 1  Receptor-mediated active targeting for the treatment of gliomas.**

| Target markers | Ligands | Nanocarriers | Loaded drugs | Ref. |
|---|---|---|---|---|
| TfR | T7 peptide | HDL | DOX | *Cui et al. (2018)* |
|  | cTfRL/a-TfR | Polymer | a-CTLA-4/a-PD-1 | *Galstyan et al. (2019)* |
| LDLR | ApoE | Micelle | paclitaxel (PTX) | *Zhang et al. (2021a)*; *Zhang et al. (2021b)* |
|  | Peptide-22 | liposome | DOX | *Chen et al. (2017)* |
|  | AP-2 | AuNP | DOX | *Ruan et al. (2015)* |
| EGFR/EGFRvIII | D−AE peptide | Micelle | PTX | *Mao et al. (2017)* |
| IL-6 receptor | I6P7 peptide | Polymer | pDNA | *Wang et al. (2017a)*; *Wang et al. (2017b)* |
| IL-13 $\alpha$2 receptor | IL-13 | Polymer | DOX | *Tu et al. (2016)* |
| nAChR | D−CDX | liposome | DOX | *Wei et al. (2015)* |
|  | D8 peptide | NLC | Bortezomib | *Farshbaf et al. (2022)* |
| Integrin protein | iRGD | Polymer | siRNA | *Gregory et al. (2020)* |
|  | c(RGDyK)/c(RGDfK) | liposome | DOX | *Belhadj et al. (2017)*; *Chen et al. (2017)* |
| VEGFR | dA7R | liposome | DOX | *Ying et al. (2018)* |
| GRP78 protein | VAP | NLC | Bortezomib | *Farshbaf et al. (2022)* |
| Folate receptor | Folic acid | CSP | DOX | *Elechalawar et al. (2019)* |
| Dopamine receptor | p-HA | liposome | DOX | *Belhadj et al. (2017)* |
| CD13 | NGR | NLC | DHA | *Zheng et al. (2013)* |

## Cytokine-mediated active targeting

Cytokines are a class of small molecule proteins with a wide range of biological activities including interleukins, interferons, tumor necrosis factor superfamily, colony-stimulating factors, chemokines, growth factors, *etc.* (*Liu et al., 2021*; *Propper & Balkwill, 2022*). Cytokines frequently used as modifying ligands are IL-6 and IL-13 (*Oh et al., 1998*; *Tu et al., 2016*). IL-13R $\alpha$2 was shown to be overexpressed in glioma cells but not in normal tissues, making it a promising target for glioma treatment (*Brown et al., 2016*; *Kim et al., 2020*; *Kim et al., 2021*).

## Amino acid-mediated active targeting

Angiopep-2 (AP-2) is a synthetic ligand for LRP-1 that is highly expressed on glioma cells (*Belhadj et al., 2017*). VAP is a stable peptide composed of 7D-amino acids that can bind to glioma cells *via* the neovascular overexpression of cell surface glucose-regulated protein 78 (csGRP78) (*Liu, Tsung & Attenello, 2020*). The use of these ligands as a conduit to deliver nanodrugs to the BBB seems promising, especially *via* the RMT pathway.

In a study, the investigators designed a nanoparticle, pV-Lip/cNC using VAP as a ligand by linking pHA to VAP using 6-aminohexanoicacid (Ahx) and modifying it onto Cabazitaxel-nanocrystals wrapped by lipid membranes, which could smoothly cross the BBB and target tumor regions (*Wu et al., 2022*).

## Small-molecule-mediated active targeting

Folic acid is the most widely used small molecule to functionalize nanoparticles, thus constituting an active targeting drug delivery platform. Folate receptors are expressed on both glioma cells and TAMs (*Fukui et al., 2020*). They play an important role in

the tumor-suppressive immune microenvironment (*Hambardzumyan, Gutmann & Kettenmann, 2016*). A CSP was designed to target glioma cells by binding to the folic acid-cationic lipid conjugate (F8) (*Elechalawar et al., 2019*). This surface-modified carbon nanocarrier loaded with DOX could effectively deliver the drug to the tumor region.

Another small molecule ligand, p-Hydroxybenzoic acid (p-HA), is a ligand for dopamine receptors and sigma receptors, both of which are present on the BBB. Since sigma receptors are overexpressed on glioma cells, pHA can be used as active targeted drug delivery (*Guo et al., 2020*; *Lu et al., 2022*). The liposome c(RGDyK)-pHA-PEG-DSPE together with pHA and liposomes with a polyethylene glycol (PEG) coating were shown to prevent liposome uptake by the reticuloendothelial system (RES) cells (*Yang et al., 2017*). They could be mounted with DOX to cross the BBB and target the tumor.

## Cell-mediated active targeting

The modification of nanoparticles with cellular components can effectively improve permeability and tumor targeting. Currently, the use of components from mesenchymal stem cells (MSCs) (*Nowak et al., 2021*), plasma component cells (*Hallal et al., 2020*), cancer cells (CCs) (*Chen et al., 2021*) and plant-derived cells (*Karamanidou & Tsouknidas, 2021*) have been validated.

A study modified NGR onto DHA-filled NLCs to achieve BBB penetration and tumor targeting by wrapping C6 CC membranes (*Chen et al., 2021*). It was shown that the immune escape and tumor homologous adhesion ability of CC membrane-encapsulated NLCs could overcome the targeting immunogenicity problem.

## BBB/glioma dual-target-mediated active targeting nanoparticles

The ability to cross the BBB to target glioma *via* dual ligand modification is an important step in an actively targeted drug delivery platform (shown in Table 2).

For example, *Farshbaf et al. (2022)* achieved promising results *in vitro* and *in vivo* in glioma models using D8/RI-VAP-modified NLCs harboring bortezomib shown in Fig. 4B. Since D8 is a ligand of nAChRs and RI-VAP has a high affinity to csGRP78, their implemented dual-modification nanoparticle could actively target glioma cell regions and BBB. As presented in Figs. 4C and 4D, both c(RGDyK)-pHA-PEG-DSPE (*Belhadj et al., 2017*) and pV-Lip/cNC (*Wu et al., 2022*) demonstrate the dual-targeting mode of the synthetic platform as well.

## *In vitro* magnetic field targeting

In addition to the active targeting, using magnetic field could be an excellent design to improve the targeting efficient of the NPs. A multifunctional nanoparticle as magnetic graphene oxide-based nanocarrier (NGO/SPION/PLGA) was used as a drug delivery platform (*Shirvalilou et al., 2018*) and was shown to effectively pass through the BBB. NGO aggregation ability has improved due to its hydrophobic surface and charge properties. At the same time, Poly (lactic-co-glycolic acid) copolymer (PLGA) can increase the biostability to prolong the circulation time (*Su et al., 2021*) and the magnetic properties of superparamagnetic iron oxide nanoparticles (SPIONs) (*Jia et al., 2018*).

**Table 2  BBB/glioma dual targeting nano drug delivery platforms.**

| Carriers | Loaded drugs | Targeting marker | Ligand | Targeting marker | Ligand | Ref. |
|---|---|---|---|---|---|---|
| Liposome | DOX | LDLR (BBB) | Pep-22 | Integrin peptide (glioma) | c(RGDfK) | *Chen et al. (2017)* |
| Liposome | DOX | Dopamine receptor (BBB) | pHA | Integrin peptide (glioma) | c(RGDyK) | *Belhadj et al. (2017)* |
| Liposome | Bortezomib | nAChR (BBB) | D8-peptide | csGRP78 (glioma) | VAP peptide | *Farshbaf et al. (2022)* |
| HDL | HCPT | TfR (BBB) | T7-peptide | VEGFR 2 (glioma) | $^d$A7R | *Cui et al. (2018)* |
| Nanocrystal | Cabazitaxel | Dopamine receptor (BBB) | pHA | Dopamine receptor/GRP78 (glioma) | pHA/VAP peptide | *Wu et al. (2022)* |
| Micelle | PTX | EGFR (BBB) | D-AE | EGFR/EGFRvIII (glioma) | D-AE | *Mao et al. (2017)* |
| Micelle | DNA | LDLR (BBB) | AP-2 | LDLR (glioma) | AP-2 | *Jiao et al. (2019)* |
| Nanogel | DOX | LRP-1 (BBB) | Lf | CD44 (glioma) | Hyaluronic acid (HA) | *Zhang et al. (2021a)*; *Zhang et al. (2021b)* |
| RBC-SLN | VCR | TfR (BBB) | T7-peptide | CD13 (glioma) | NGR | *Fu et al. (2019)*; *Fu et al. (2019b)* |
| BSA-NP | DOX | LfR (BBB) | Lf | LfR/LRP (glioma) | Lf | *Su et al. (2014)* |
| AuNP | DOX | $\alpha v \beta 3$ receptor (BBB) | R8-RGD | $\alpha v \beta 3$ receptor (glioma) | R8-RGD | *Ruan et al. (2017)* |
| SPION | Cy5.5 | Integrin peptide (BBB) | c(RGDyK) | Folate receptor (glioma) | FA | *Zhang et al. (2016)* |
| SPION | DOX | LfR (BBB) | Lf | LfR (glioma) | Lf | *Fang et al. (2014)* |
| CND | DOX | IL-6R (BBB) | $I_6P_8$-peptide | IL-6R (glioma) | $I_6P_8$-peptide | *Wang et al. (2017a)*; *Wang et al. (2017b)* |
| Graphene | DOX | LfR (BBB) | Lf | LfR (glioma) | Lf | *Song et al. (2017)* |

**Notes.**

VCR, vinca alkaloid vincristine.; HCPT, 10-hydroxycamptothecin.

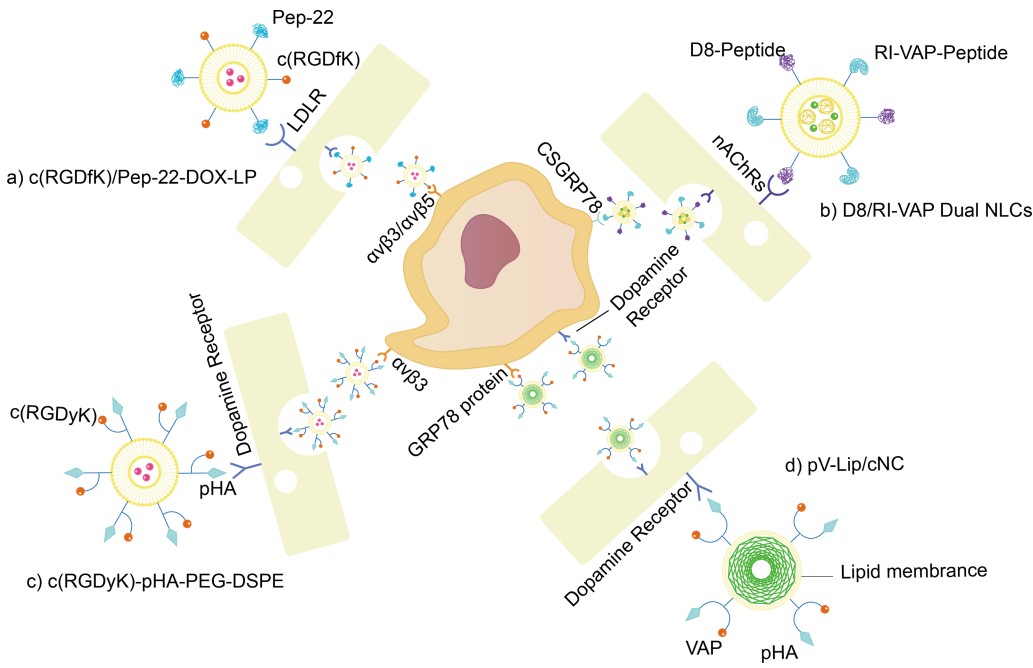

**Figure 4** **BBB/glioma dual targeting nano drug delivery platform.** (A) c(RGDfK)/Pep-22-DOX-LP. Liposomes with Pep-22 and c(RGDfK) dual modified are loaded with DOX. Pep-22 targets LDLR expressed on BBB, while c(RGDfK) targets integrin protein overexpressed on gliomas. (B) D8/RI-VAP Dual NLCs.NLC with RI-VAP-peptide and D8-peptide dual modified are loaded with bortezomib. D8-peptide targets nAChRs expressed on BBB, while RI-VAP-peptide targets csGRP78 overexpressed on gliomas. (C) c(RGDyK)-pHA-PEG-DSPE. Liposomes with pHA and c(RGDyK) dual modified are loaded with DOX. pHA targets dopamine receptors expressed on BBB, while c(RGDyK) targets integrin protein overexpressed on gliomas. (D) pV-Lip/cNC. Nanocrystals loaded with Cabazitaxel, are modified with pHA and VAP-peptide, coated by lipid membrane. pHA targets dopamine receptors expressed on both BBB and gliomas, while VAP-peptide targets csGRP78 overexpressed on gliomas.

## CONCLUSION AND PERSPECTIVE

The application of nanomaterials for spanning BBB and targeted drug delivery to gliomas is a great breakthrough in the field of drug therapy for gliomas. This article provided some examples of different nano drug delivery platforms that have been proposed for targeting gliomas. It is evident from the results of these studies that nano-delivery platforms could greatly increase the possibility of effective drug therapy for gliomas in the near future. In addition, the fusion application of multiple nanodrug delivery platforms might also overcome the shortcomings of single nanocarriers and maximize the advantages of nanocarriers of different materials. Further, despite the functionalized modification of nanoparticles has been extensively studied in recent years to improve their ability to cross the BBB (*Pourgholi et al., 2016*), but there are still some challenges including (*Ferraris et al., 2020*; *Karlsson et al., 2021*) (1) stability: NPs can be unstable *in vivo*, leading to premature release of drugs and reducing the efficacy of treatment. New materials and

formulations need to be developed to improve the stability; (2) toxicity: Some types of NPs can be toxic to healthy cells. The development of new materials and coatings, as well as improved targeting, and improved biocompatibility, clearance, *etc.* may be the way to address this issue; (3) clinical translation: nanoparticles drug delivery is still in the early stages of clinical development. Much further work needs to be achieved, such as safety and efficacy in humans, development of new imaging techniques to monitor the distribution and uptake of NPs in the brain, and improved therapeutic outcomes; (4) personalized medicine: gliomas are a heterogeneous group of tumors, and different patients may respond differently to nanoparticles drug delivery, which may require the use of imaging techniques to identify the molecular and genetic signatures of individual tumors, and developing nanoparticles that can selectively target these specific tumor features; (5) combination therapy: combining nanoparticles drug delivery with other treatment modalities, such as radiation therapy, chemotherapy, or immunotherapy, may improve treatment outcomes for glioma patients and to maximize synergistic effects of combination therapy as well as outcome assessment.

The current research is gradually solving the problems of permeability, stability and toxicity of NPs, and has made considerable achievements, but as for the clinical translation and personalized treatment, it still needs time to be realized.

### Funding

This work was supported by grants from the Shaoxing Municipal Science and Technology Plan Project of China Under Grant (2022A14022), the Shaoxing Health Science and Technology Project of China Under Grant (2022KY003) and the Shaoxing Health Science and Technology Project of China Under Grant (2022KY016). The funders had no role in study design, data collection and analysis, decision to publish, or preparation of the manuscript.

### Grant Disclosures

The following grant information was disclosed by the authors:
Shaoxing Municipal Science and Technology Plan Project of China: 2022A14022.
Shaoxing Health Science and Technology Project of China: 2022KY003, 2022KY016.

### Competing Interests

The authors declare there are no competing interests.

### Author Contributions

- Yongyan Wu conceived and designed the experiments, performed the experiments, analyzed the data, prepared figures and/or tables, authored or reviewed drafts of the article, and approved the final draft.
- Yufeng Qian conceived and designed the experiments, performed the experiments, analyzed the data, prepared figures and/or tables, authored or reviewed drafts of the article, and approved the final draft.
- Wei Peng conceived and designed the experiments, performed the experiments, analyzed the data, prepared figures and/or tables, authored or reviewed drafts of the article, and approved the final draft.
- Xuchen Qi conceived and designed the experiments, performed the experiments, analyzed the data, prepared figures and/or tables, authored or reviewed drafts of the article, and approved the final draft.

## Data Availability

This is a literature review.

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
