# Peer review of "Functionalized nanoparticles crossing the brain–blood barrier to target glioma cells"

_PeerJ, doi:10.7717/peerj.15571_

## Round 0.1 · original submission · Minor Revisions

Please carefully read the comments and address the questions.

Reviewer 1 ·

Basic reporting

Clear and unambiguous, professional English used throughout this review. Literature references, sufficient field background provided. Professional article structure, wonderful figures and tables. The Introduction adequately introduce the subject and make it clear who the audience is/what the motivation is.

Experimental design

Article content is within the Aims and Scope of the journal. Rigorous investigation performed to a high technical & ethical standard. All sources adequately cited. Quoted as appropriate. This review is organized logically into coherent subsections.

Validity of the findings

Conclusions are well stated, linked to original research question & limited to supporting results. The Conclusion identify unresolved questions such as the clinical application of nanocarriers is still
far away.

·

Basic reporting

There are spelling errors in the text, and it is recommended to make corrections. For example, in line 388, np should be capitalized.

Experimental design

no comment

Validity of the findings

no comment

Additional comments

1. The author introduces the common receptors overexpressed on the surface of glioma cell membranes in line 172 of the article. However, there are too few types, and it is recommended to supplement other glioma specific expression receptors. For example, Lactoferrin, Folic acid targeting Folate acceptor, etc.
2. It is recommended that the author provide examples of which ligands are applied to target binding LDLR in section 2.2.
3. It is recommended that the author provide an introduction in the text about Tumor microenvironment responsive nanoparticles and Nanoparticles modified with specific cell membrane surfaces.

Reviewer 3 ·

Basic reporting

Wu et al. the characteristics and pathways of different nanocarriers for crossing the BBB and targeting glioma by listing different materials for drug delivery platforms. The article is clear and logical. However, there are some issues that prevent the article from being published in its current form. Here are my suggestions:
1. The writer needs to double-check the grammar, spelling and punctuation of the whole article.
2. The essay needs a flow chart.

Experimental design

no comment

Validity of the findings

no comment

Additional comments

The main problem:
1. Figure 2 shows nanoparticles with functionalizing modification and drug loading, but it seems need to provide a detailed explanation of the specific meaning explained in the chart.
2. Table 3 shows the nano drug delivery platforms, but only list three Organic nano carries, more other carries should also be included in such as graphene, quantum dots, SPION, golden NPs, etc. The author should refer to and cite more related research articles.
3. Drug loaded nano delivery systems have become one of the most promising methods for treating glioma. The drug loaded nano delivery system has advantages such as high drug loading and long blood circulation time, but it also has certain shortcomings. The author should indicate the current research bottlenecks and future research directions.
4. The author's references are not very comprehensive. The authors forget to cite as evidence similar articles using the same analytical tools.

Reviewer 4 ·

Basic reporting

The review highlights the use of the functionalized nanoparticles to cross the blood-brain barrier. The review does a decent job of discussing major hurdles in crossing the blood-brain barrier and the different classes of nanoparticles. The manuscript could be improved with further expansion of nanoparticles use specifically in treating gliomas. The introduction discusses the challenges of crossing the barrier and then modestly discusses the differences between an intact BBB and a leakier blood-brain-tumor barrier. This could be expanded upon by discussing differences in barrier phenotypes (i.e. tight junction dysfunction, altered efflux transporter, etc.) between the two barrier "types" in glioma; an additional figure could be helpful here. Furthermore, the review briefly mentions efflux transporter's role in an active barrier. Efflux transporters are one of if not the greatest hurdle when it comes to crossing the barrier and this review did not adequately address those challenges. It is unclear how " drug molecules and their modifiers can also use these efflux-mediated proteins to cross the BBB" as this was not expanded upon in the review.

Experimental design

The review appears to be unbiased and covers the material well. An additional figure highlighting the effects of glioma on the BBB and the critical differences between healthy and glioma barriers would be helpful. Figure 1 is a little confusing with the location of efflux transporters and could be improved.

Validity of the findings

No Comment

Additional comments

Please italicize in vitro and in vivo throughout the manuscript.
Please alternate gray and white in your tables to distinguish the separate rows.

---

## Round 0.2 · accepted · Accept

The authors have addressed all the comments from reviewers.